# Applying global warming levels of emergence to highlight the increasing population exposure to temperature and precipitation extremes

5 David Gampe[1], Clemens Schwingshackl[1], Andrea Böhnisch[1], Magdalena Mittermeier[1], Marit Sandstad[2], Raul R. Wood[1,3,4]

[1] Dept. of Geography, Ludwig-Maximilians-Universität München, Munich, Germany
[2] CICERO Center for International Climate Research, Oslo, Norway
[3] WSL Institute for Snow and Avalanche Research SLF, Davos Dorf, Switzerland
10 [4] Climate Change, Extremes and Natural Hazards in Alpine Regions Research Center CERC, Davos Dorf, Switzerland

Correspondence to: David Gampe (d.gampe@lmu.de)

**Abstract.** The swift and ongoing rise of global temperatures over the past decades led to an increasing number of climate variables showing statistically significant changes compared to their pre-industrial state. Determining when these climate signals emerge from the noise of internal climate variability (i.e., estimating the Time of Emergence, ToE) is crucial for climate risk assessments and adaptation planning. However, robustly disentangling the climate signal from internal variability represents a challenging task. While climate projections are communicated increasingly frequently through global warming levels (GWLs), ToE is usually still expressed in terms of time horizons. Here, we present a framework to robustly derive global warming levels of emergence (GWLoE) using five Single Model Initial-condition Large Ensembles (SMILEs) and apply it to four selected temperature and precipitation indices. We show that the concept of GWLoE is particularly promising to constrain temperature projections and proves a viable tool to communicate scientific results. We find that >85% of the global population is exposed to emerged signals of nighttime temperatures at a GWL of 1.5°C, increasing to >95% at 2.0°C. Daily maximum temperature follows a similar, yet less pronounced path. Emerged signals for mean and extreme precipitation start appearing at current GWLs and increase steadily with further warming (~10% population exposed at 2.0°C). Related probability ratios for the occurrence of extremes indicate a strong increase with widespread saturation of temperature extremes (extremes relative to historical conditions occur every year) reached below 2.5°C warming particularly in (sub)tropical regions. These results indicate that we are in a critical period for climate action as every fraction of additional warming substantially increases the adverse effects on human wellbeing.

## 1 Introduction

The sixth Assessment Report of the Intergovernmental Panel on Climate Change (IPCC) repeatedly confirmed that the recent global warming is unequivocally caused by anthropogenic activity (Masson-Delmotte et al. 2021). The latest decade (2011-2020) saw 1.1°C higher global temperatures compared to pre-industrial times (1850-1900) and warming is projected to continue in the future under current climate policies (IPCC 2022). To prevent adverse and potentially catastrophic impacts of very high warming rates, the Paris Agreement urges to hold global warming "well below 2.0°C above pre-industrial levels", ideally limiting it to 1.5°C (UNFCCC, 2015). However, a warming of 1.5°C will already impose negative impacts on ecosystems and human wellbeing (Masson-Delmotte et al. 2018), and a growing body of literature highlights the adverse consequences of even higher warming rates (e.g., Hoegh-Guldberg 2019, Schwingshackl et al. 2021). Many studies have elaborated the benefits of limiting global warming to 1.5°C compared to 2°C. These studies show, among others, substantially less area affected by desertification (Park et al. 2018), less population exposed to extreme daily temperatures (Harrington 2021, King & Karoly 2017), a lower reduction in water availability and a smaller increase in dry spell length (Schleussner et al. 2016), as well as a less pronounced increase in drought risk and risk of consecutive drought years (Lehner et al. 2017a). Given the current warming rate and the expected severe impacts if exceeding 1.5°C of warming, it is essential to estimate the

consequences of warming levels beyond political targets at incremental steps. Hence, it is important to analyze at which warming level we can expect a significant signal to emerge. Incremental steps of GWL are detrimental, since a discernable response to even strong and sustained mitigation can be delayed by decades due to the inertia and internal variability of the climate system (Samset et al. 2020).

The time of emergence (ToE) is a well-established concept to estimate whether and when a climate change signal is detectable (e.g., Lehner et al. 2017b, Hawkins and Sutton 2012). ToE indicates the time when the considered climate variable changes into a new state. This is generally estimated by testing whether the distribution of this variable is statistically significantly different from the respective distribution in a world without climate change. While expressing ToE as distinct years is illustrative and easy to communicate, uncertainties of climate projections make a precise estimation challenging (Hawkins et al. 2014). Climate projections are subject to three major sources of uncertainty: uncertainty due to internal variability of the climate system, structural uncertainty introduced by different model parameterizations, and scenario uncertainty reflecting differences in potential future socioeconomic and related emission pathways (Hawkins & Sutton, 2009; Lehner et al. 2020). Various methods have been developed to quantify, distinguish and constrain the different types of uncertainty (Lehner et al. 2023).

Most ToE studies use multi-model ensembles, which mainly consist of single realizations of different models, thus accounting for structural uncertainty and scenario uncertainty (Giorgi and Bi 2009, King et al. 2015, Bador et al 2016, Douglas et al. 2022). However, these single realization ensembles can only partially account for the intrinsic uncertainty due to internal climate variability. Especially on regional-to-local scales, internal variability is large compared to the other sources of uncertainty, showing the largest fractional uncertainty (Lehner et al. 2020, Blanusa et al. 2023). Accounting for internal variability when estimating ToE, is relevant since it can advance or delay ToE by up to several decades (Hawkins et al 2014), and can contribute half to two-thirds to the total ToE uncertainty (Bador et al 2016). To account for the influence of internal variability in ToE studies, Single Model initial condition large ensembles (SMILEs) can be applied. SMILEs constitute numerous independent, yet equally probable, climate simulations, created by running a single climate model multiple times under the same external forcing (e.g., same emission scenario) but with marginally changed initial conditions (Kay et al. 2015, Maher et al. 2019). It has been shown that SMILEs are ideal tools to estimate ToE due to their ability to provide both statistically robust forced signals and accurate quantifications of internal variability via the spread across ensemble members (Lehner et al. 2017b, Schlunegger et al. 2019, 2020, Wood and Ludwig 2020). SMILEs are widely used and have been proven to effectively disentangle a robust forced response from internal variability (e.g., Deser et al. 2020, Maher et al. 2021). Further, they represent a valuable tool for a robust assessment of extremes by extensively sampling the tails of the distribution (Suarez-Gutierrez et al. 2020, Wood et al. 2021). The increasing number and availability of SMILEs over recent years (Deser et al. 2020) makes it possible to additionally address structural uncertainty. Merging the information of multiple SMILEs to assess the corresponding joint time of emergence should thus allow for an even more robust detection of ToE, as internal variability and model uncertainty can both be assessed.

In recent years, future climate projections have been expressed increasingly frequently through global warming levels (GWLs) instead of fixed time horizons (e.g., the period 2071-2100; Seneviratne et al. 2021). This approach constrains scenario uncertainty by the question of which GWL will be reached and expresses future climate projections in a more policy-relevant way. Recently, first studies combined GWL and ToE to provide global warming levels of emergence (GWLoE) instead of ToE in single realization multi-model ensembles (Abatzoglou et al. 2019, Seneviratne & Hauser 2020), reanalysis and observations (Raymond et al. 2020), and in two SMILEs (Kirchmeier-Young et al. 2019). Yet, GWLoE remains a rarely applied concept, particularly an application using multiple SMILEs in a joint emergence framework is lacking. In this study, we thus quantify the joint GWLoE of selected temperature and precipitation indices using multiple SMILEs from the Coupled Model Intercomparison Project Phase 6 (CMIP6). By using an ensemble of multiple SMILEs, we are able to robustly determine the emergence as a function of GWL at the grid scale level, implicitly accounting for internal variability and structural uncertainty. Expressing emergence as GWLs instead of time thereby constrains the scenario uncertainty. Further, we relate incremental changes in GWL to changes in the exposure to temperature and precipitation extremes by estimating increases in their probability ratios for each 0.1°C warming to analyze the linear or non-linear response to warming. Lastly, we quantify the exposure of population and land area, on a global and regional scale, to emerged climate indices as a function of GWL.

## 2 Materials and Methods

### 2.1 SMILEs and climate indices

We use five different SMILEs that are part of the CMIP6 archive (ACCESS-ESM1-5, CanESM5, EC-Earth3, MIROC6, and MPI-ESM1-2LR; see Tab. 1) with a sufficient number of ensemble members (30-50) to represent internal climate variability (Milinski et al. 2020, Tebaldi et al. 2021). A sufficiently large ensemble size is particularly relevant for precipitation variability, where the ensemble should comprise at least 30 members (Wood et al. 2021). We selected four temperature and precipitation indices compiled by the Expert Team on Climate Change Detection and Indices (ETCCDI): yearly maximum of daily maximum temperature (TXx), yearly maximum of daily minimum temperature (TNx), total annual precipitation (PRCPtot), and yearly maximum 1-day precipitation (Rx1day). We selected those four indices as they are frequently applied (e.g. Sillmann et al. 2013, Deng et al. 2022) and allow for easy interpretability. We further aim to demonstrate the concept of GWLoE for a broad range of indices. The selected temperature and precipitation indices thus intentionally cover both, absolute (TXx, TNx and PRCPtot) and intensity (Rx1day) metrics.

To make the results comparable across SMILEs and to further calculate the joint emergence using multiple SMILEs, the grids of the five SMILEs must be harmonized. Typically, either the finest or coarsest grid is selected as target resolution. The selection of the finest grid exploits the potential of the high-resolution models. The coarser models, however, might not be capable of resolving the processes at the higher resolution for structural and parameterizational reasons (Prein et al. 2016). Using the finest grid would then also require the introduction of new data points (either through interpolation or downscaling).

We thus opted to remap all data sets to the spatial resolution of the coarsest grid (CanESM5, ~2.8°x2.8°; Tab. 1) using a first order conservative remapping approach.

We further aim at analyzing a wide range of potential future GWLs to identify the impact of incremental changes of global warming on selected indices and the related emerging risks. Hence, we selected SMILEs under historical forcing and the high-end scenario SSP5-8.5, which projects an increase in radiative forcing of 8.5 W/m$^2$ by the end of the 21$^{st}$ century (Gidden et al. 2019). The choice of this rather high-end scenario allows us to analyze high warming levels (above 3.5°C) compared to pre-industrial conditions (1850-1900; Fig. 1). In contrast, some of the lower emission scenarios might not even reach GWLs of 1.5°C to 2°C by the end of the century despite an observed global warming of already more than 1.1°C over the last decade (2011-2020; Fig. 1). Overall, the range of GWLs projected by the five selected SMILEs for the end of the 21$^{st}$ century (3.8°C – 7.1°C; Fig. 1) is in general agreement with the respective spread of the full CMIP6 ensemble (Tebaldi et al. 2021).

**Table 1: Overview of the five Single Model Initial-condition Large Ensembles (SMILEs) applied in this study. The CMIP6 historical and SSP5-8.5 scenarios (in total covering the period 1850-2100) were used for all SMILEs. All models were conservatively remapped to the coarsest model grid (CanESM5) for further analysis. The values for Equilibrium Climate Sensitivity (ECS) stem from Meehl et al. (2020) and provide an estimate of the climate sensitivity of each SMILE.**

| SMILE | Ensemble size (n members) | Original resolution (lat x lon grid) | ECS (°C) | Reference |
|---|---|---|---|---|
| ACCESS-ESM1-5 | 40 | ~1.3°x1.9° | 3.9 | Mackallah et al. 2022 |
| CanESM5 | 50 | ~2.8°x2.8° | 5.6 | Swart et al. 2019 |
| EC-Earth3 | 50 | ~0.7°x0.7° | 4.3 | Wyser et al. 2021 |
| MIROC6 | 50 | ~1.4°x1.4° | 2.6 | Tatebe et al. 2019 |
| MPI-ESM1-2LR | 30 | ~1.9°x1.9° | 3.0 | Mauritsen et al. 2019 |

**2.2 Time of Emergence (ToE) and Global Warming Level of Emergence (GWLoE)**

To calculate ToE, we extract 20-year moving windows for each year over the period 1901 to 2100. A two-sided Kolmogorov-Smirnov test (KS-test) at 5% significance level is then used to test the resemblance to the pre-industrial reference climate state (1850-1900; Mahlstein et al. 2012, King et al. 2015). The climate signal is considered emerged once the KS-test indicates that a 20-year time series was drawn from a different distribution than the reference data. These differences may refer to distribution shifts (mean changes) and shape changes (variability). It is important to account for both changes, because they can either individually or collectively attribute to the changes in extremes (van der Wiel and Bintanja 2021, Wood 2023). For each ensemble member, we define ToE as the tenth year of the first 20-year window where the p-value of the KS-test determines significance. We further require that the KS-test remains significant in all subsequent periods as well. The climate signal is

considered as not emerged by the end of the 21$^{st}$ century if the KS-test for the last 20-year window (2081-2100) does not yield significant differences. The calculations are carried out for each index and each SMILE member on the grid cell level.

To estimate the sampling uncertainty in the calculation of the emergence, a bootstrapping approach is conducted. We sample $n$ members ($n$ = ensemble size) of each SMILE from the available members with replacement, applying 1000 repetitions. Thereby, all individual members were sampled approximately equal times. For each of these 1000 bootstrapped ensembles, we then assign ToE to the year when at least 90% of the drawn ensemble members show an emerged climate signal (e.g., 45 of 50 members; similar to the approach by Martel et al. 2018). This method accounts for internal variability, expressed via the $n$ SMILE members. As we require 90% of the members to be emerged, the approach yields a conservative yet robust estimate of ToE. The sampling uncertainty is presented as the 95% confidence interval derived from the 1000 ToE estimates. To transfer ToE into Global Warming Level of Emergence (GWLoE), we calculate GWL as the change in the area-weighted global average of annual surface air temperature (GSAT) in each moving 20-year window relative to the pre-industrial period following the approach by Hauser et al. (2019) as used in IPCC AR6 (Seneviratne et al. 2021). The GSAT changes are assigned to the tenth year of each 20-year period and define the GWL for that year in each member of each SMILE. To derive GWLoE, we assign the corresponding GWL to the previously calculated year of climate signal emergence (ToE) for each of the $n$ members. Thus, replacing the time axis with GWL in all of the 1000 bootstrapped ensembles through their sampled members. The final GWLoE of a SMILE is then defined as the mean across all bootstrapped ensemble members (i.e., the forced response), and the confidence interval is obtained using the same methodology as for ToE.

To further increase the robustness of the GWLoE estimates, we calculate the joint emergence of the climate signal across all five SMILEs. We define this joint emergence as the median GWLoE of the five SMILEs, calculated for each index at the grid cell level. We additionally claim that SMILEs agree in the signal emergence if at least four out of the five SMILEs indicate an emergence within the 21$^{st}$ century. Finally, we cap the GWLoE at 4°C as not all SMILEs reach that warming level by 2100 (Fig. 1), any emergence after 4°C is considered not emerged.

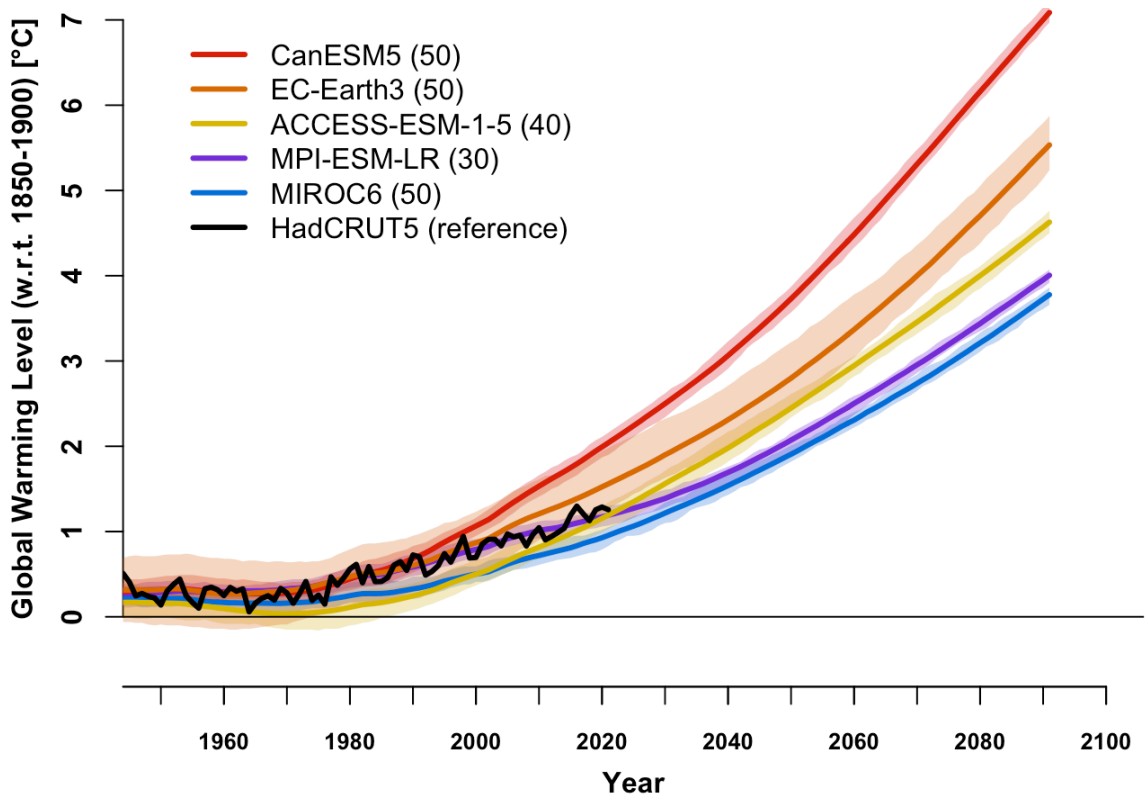

**Figure 1: Changes in global average annual surface air temperature, i.e., Global Warming Level (GWL). GWL is calculated relative to pre-industrial conditions (1850-1900 under historical and SSP5-8.5 scenarios and is presented for the five SMILEs (colors indicate their respective equilibrium climate sensitivity (ECS) from low (blue) to high (red)) and the blended, observation-based reference data set HadCRUT5 (black; Morice et al. 2021). Solid lines indicate the ensemble mean and shaded areas represent the range**

**(minimum-to-maximum) of the individual members for each SMILE. Note that following our methodology, GWL was calculated based on 20-year moving windows. The GWL shown in this figure thus also reflects 20-year moving windows and not annual data. Accordingly, the presented ensemble spread does not represent inter-annual variability and is considerably narrower than for non-smoothed, annual data. Numbers in the legend indicate the ensemble size of each SMILE (*n* members).**

## 2.3 Exposure of Population and Land Area to Emerged Climate Signals

For each of the four climate indices, we quantify the fraction of population and the land area fraction affected by emerged climate signals. We use historical population data from ISIMIP2b (Frieler et al., 2017) and future population scenarios according to the different SSPs (SSP1-SSP5; Jones & O'Neill, 2016, Samir & Lutz, 2017). We opted to include a wide range of population projections despite a potential disagreement with the selected climate scenario (SSP5-8.5) to analyze the impact of different possible population trajectories on our results, as the usage of GWLs should make the estimated emergence largely

independent of the underlying emission scenario (i.e., RCP 8.5 in our case). We calculate the population density for each population dataset and remap it to the common grid (CanESM5 grid; see Section 2.1) using first order conservative remapping. As the SSP population data are available in 10-year intervals, we interpolate them linearly in time to obtain annual data. To estimate the time-dependent population exposure to emerged climate signals, the population of all respective grid cells is

aggregated globally or to larger regions. We express the result as percentage of (time-dependent) global and regional population. Similarly, we calculate the fraction of global (and regional) land area, on which a climate signal emerges, using the (time-invariant) land area fraction of CanESM5.

As population projections are time-dependent, the population emergence as a function of GWL can only be quantified by considering the temporal evolution of GWL. For each GWL, we use the population of the year in which the GWL is reached for the first time (see Section 2.2) and calculate the global (and regional) population exposure by considering all grid cells that have emerged at that GWL and the population in the respective year. Thus, population changes in a grid cell that have already emerged at a lower GWL continue to influence the total population exposure at higher GWLs. We further quantify the uncertainty due to internal variability by estimating the exposure of land area and population individually for each of the 1000 bootstrapped ensembles and by calculating the 95% confidence interval across all members.

## 2.4 Changes in Probability Ratio of Climate Index Extremes

In addition to considering the full probability distribution of climate indices when determining ToE and GWLoE (based on KS-test), we further quantify how the probability of their extreme values changes with global warming. We define the frequency of extreme years as the number of years exceeding the 95$^{th}$ percentile of the respective climate index distribution in the reference period 1850-1900. We calculate the probability ratio $PR$ based on frequencies of extreme events for each 20-year period as

$$(1) \quad PR = \frac{\frac{n_{fut}}{y_{fut} \cdot m_{fut}}}{\frac{n_{ref}}{y_{ref} \cdot m_{ref}}}$$

where $n$ is the event frequency during the reference (*ref*) and future (*fut*) periods pooled across all members, $y$ the period length (20 years for *fut*, 51 years for *ref*) and $m$ the number of ensemble members. Probability ratios above (below) 1 indicate an increase (decrease) in event frequency relative to the reference period 1850-1900. By definition, the event probability equals 0.05 in the reference period when considering the 95$^{th}$ percentile threshold. This is equivalent to a return period of 20 years for annual maximum temperature and precipitation events (see Supplementary Fig. S1a for a conceptual illustration). Therefore, the theoretical maximum probability ratio, during any 20-year period, is $PR=20$, which indicates that pre-industrial extreme thresholds are exceeded every year in all SMILE members. We quantify the GWL at which this saturation effect occurs for each of the four indices.

Furthermore, to derive the change in probability ratios of extreme years as a function of GWL, we linearly regress the probability ratio against GWL using the least-squares approach. We account for non-linear changes in probability ratios across the considered GWL range (1°C to 4°C) by performing the linear regression piecewise for three global warming intervals: 1°C to 2°C, 2°C to 3°C, and 3°C to 4°C (see conceptual illustration in Supplementary Fig. S1b). The estimated regression coefficients indicate the magnitude of changes in probability ratios per tenth of a degree (0.1°C) of additional global warming.

To account for inter-SMILE differences, we average the regression coefficients, weighted by the number of SMILE members, and mask out areas where fewer than four SMILEs agree in the direction of the PR change.

The 0.1°C GWL step we apply is finer than the steps used by other studies to investigate frequency changes at distinct GWL thresholds (e.g., GWLs of 1.5°C or 2.0°C related to the Paris Agreement). Those studies commonly employ distinct GWLs or increments of 0.5°C or 1°C to obtain statistically robust change signals (Perkins-Kirkpatrick & Gibson 2017; King et al. 2018; Fischer & Knutti 2015). However, our setup using five SMILEs, with 30-50 ensemble members each (220 members in total), ensures robust assessments and allows us to analyze frequency changes of extreme events at incremental GWLs. By considering GWL steps of 0.1°C, we are able to evaluate the contribution of incremental warming steps to increases in extreme event frequency with a particular focus on different warming intervals. This also allows to illustrate the potential consequences of overshooting policy-agreed GWL targets by even a small margin.

## 3 Results

### 3.1 Global warming level of emergence for temperature and precipitation indices

The joint emergences of the considered indices across all SMILEs show distinct GWLoE patterns (Fig. 2). In particular, the temperature indices show a widespread emergence at low GWLs, with substantial emergence occurring at present-day GWL (around 1.1°C). This indicates that many regions have already transitioned into a new climate state for TXx and TNx. Emergences of TXx are particularly prevalent in the Southern Hemisphere, including large parts of Africa and South America, as well as Southern Europe, Central America, and the Arabian Peninsula (Fig. 2a). In all other regions, TXx is projected to emerge between a GWL of 1.0°C and 2.0°C except for a few regions with emergence only at higher GWLs. TNx shows an even more pronounced and widespread emergence at present-day warming (except for Antarctica), reflecting that climate change has already impacted the temperature indices across the globe (Fig. 2b). The model agreement for the emergence of the temperature indices is very high (no areas are hatched in Fig. 2a, b). While the joint emergence of all SMILEs provides an estimate of GWLoE based on the median GWLoE across the five SMILEs, individual models emerge at lower or higher GWLs due to internal variability. Thus, the range of GWLoE across SMILEs provides additional information on the robustness of the results. The robustness is particularly high for TNx, as indicated by a narrow range of GWLoE across SMILEs (Supplementary Fig. S2). While the range yields a generally high agreement also for TXx, the patterns are more diverse, manifested by a larger range in eastern North America, eastern Europe, Central Africa and parts of South East Asia (Supplementary Fig. S2).

The precipitation indices generally emerge less widespread and at higher GWLs than the temperature indices (Fig. 2c, d). PRCPtot emerges at a GWL of around 2°C in the Northern high latitudes, central Asia, and parts of tropical Africa and South America (Fig. 2c). For most of these regions (except for South America), a general increase in annual precipitation is projected (IPCC, 2021). Rx1day is generally projected to increase over land due to dynamical and thermodynamical adjustments (Seneviratne et al. 2021). However, the Rx1day signal only emerges in parts of Africa and South America for GWL <2.0°C

(Fig. 2d). For the rest of the globe, PRCPtot and Rx1day do not emerge until a GWL of 4°C or higher, with large areas (particularly dry regions) showing no emergence at all within the considered GWL range. In addition to high internal variability (Supplementary Fig. S15), the inter-model range in GWLoE for precipitation indices is substantially larger than for temperature indices, partly explaining that the precipitation indices only emerge at higher GWLs (Supplementary Fig. S2). Regions with a narrower GWLoE range predominantly correspond to grid cells where the signals emerged in fewer than four SMILEs. The

narrow range in these regions is thus due to fewer SMILEs and does not necessarily indicate increased robustness.

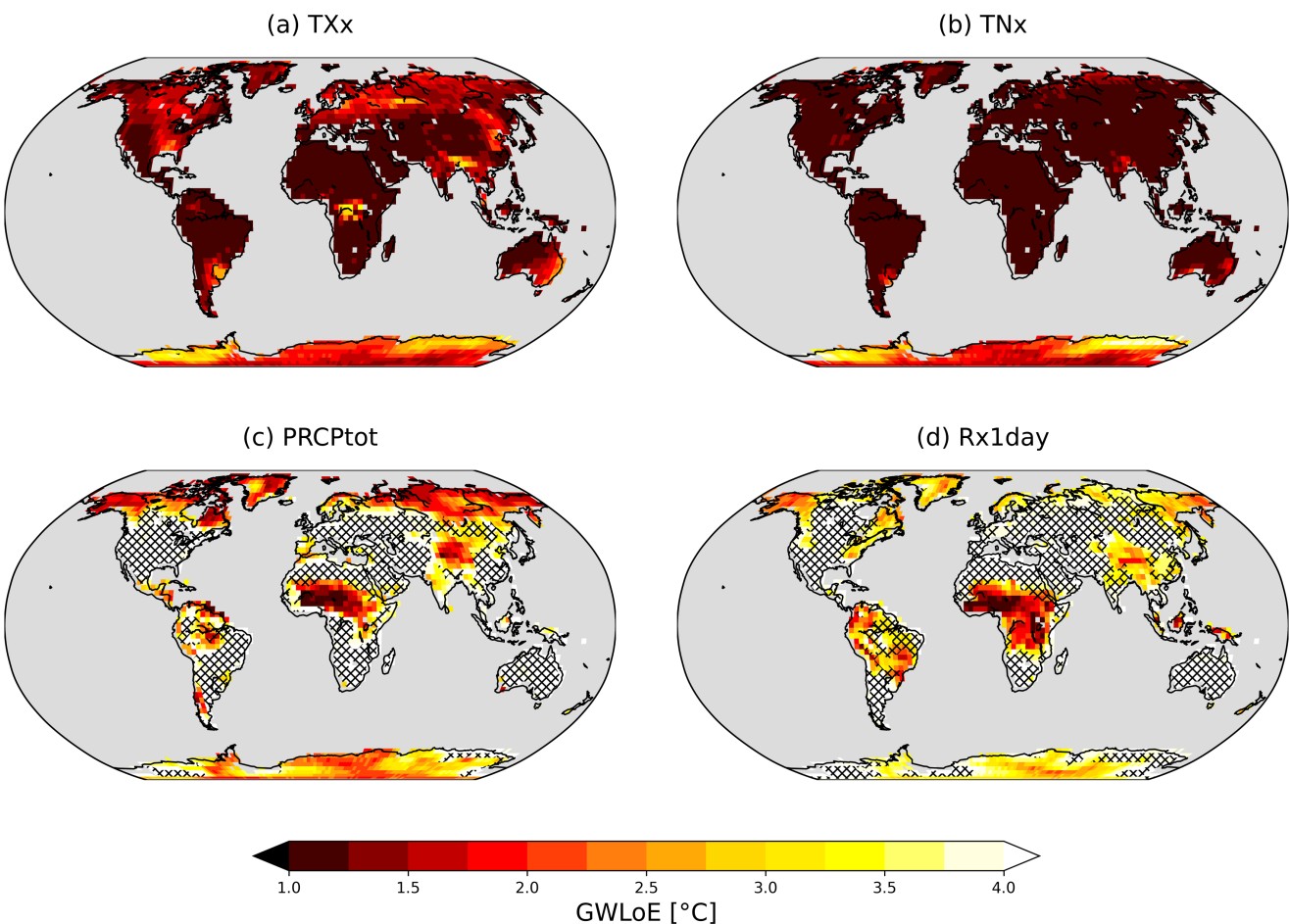

**Figure 2: Joint Global Warming Level of Emergence (GWLoE) of the considered indices.** Maps show the joint emergence (multi-model median) of the five applied bootstrapped SMILEs (see Methods) using historical and SSP5-8.5 scenarios for TXx (a), TNx (b),

PRCPtot (c), and Rx1day (d). Red colors indicate an earlier emergence. Hatched areas indicate regions where emergence within the considered GWL range is detected in fewer than four of the five SMILEs. Grid cells that do not yield emergence at GWL<4°C are colored white. Light grey indicates non-land grid cells.

## 3.2. Exposure of land area and population to emerged climate signals

To quantify how the spatial extent of the emerged climate signals changes over time, we next assess the percentage of land area exposed to emerged climate signals as a function of GWL (Fig. 3). TXx has already emerged on 41%-56% (range across all SMILEs) of the global land area under present-day climate. The emergence continues to increase linearly until stabilizing around 2°C when most of the land fraction shows emerged signals (78%-87%; Fig. 3a). Africa, West & South Asia, South America, and Southeast Asia can be identified as hotspots where TXx has emerged on most of the land already under present-day climate (Supplementary Fig. S3). TNx shows a similar path, but with a larger fraction of land area having emerged climate signals under present-day conditions (53%-83%; Fig. 3b and Supplementary Fig. S3). At a GWL of 1.5°C, 86%-94% of the land area will be exposed to a new climate state for TNx (Fig. 3b). All five SMILEs show similar changes for TXx and TNx globally and for most of the analyzed regions, except for Southeast Asia and Australia, where the land area fraction with emerged signals varies strongly across the different SMILEs (Supplementary Fig. S3).

The emergence of climate signals for the precipitation indices occurs at higher GWLs than for the temperature indices. Emergences are thus only detectable over a small portion of the land area under present-day climate (1%-18%; Fig. 3c, d). The fraction of land exposed to emergences of PRCPtot shows a linear increase from around 1.5°C onwards, with roughly a fifth (11%-37%) of the land area being exposed to a new climate state at a GWL of 2.0°C (Fig. 3c). However, we find strong regional differences, with emerged signals at 2.0°C warming being more widespread in North & Central Asia, Africa, and South America than on global average (Supplementary Fig. S3). Particularly in North & Central Asia, the estimated exposed land fraction also shows substantial differences across the five SMILEs. The fraction of land exposed to Rx1day emergences increases linearly as well, starting at a GWL of around 1.0°C, but the rate of increase depends strongly on the considered SMILE. Three of the five SMILEs (MPI-ESM1-2-LR, MIROC6, and ACCESS-ESM1-5) follow a similar path (around 8%-10% of exposed area at a GWL of 2.0°C), while EC-Earth3 shows a much higher exposed area (43%) and CanESM5 a much lower exposed area (2%) at 2.0°C.

The related sampling uncertainty due to composition of the ensemble members at global scale (shaded area in Figure 3; 95% confidence interval from bootstrapped member sampling) is very low for all four climate indices. This is also the case on the regional scale (Supplementary Figure S3) except for Southeast Asia and Australia, where the sampling uncertainty plays a larger role than in the other regions. Nevertheless, the structural uncertainty, originating from differences across SMILEs, constitutes the dominating source of uncertainty, both globally and regionally.

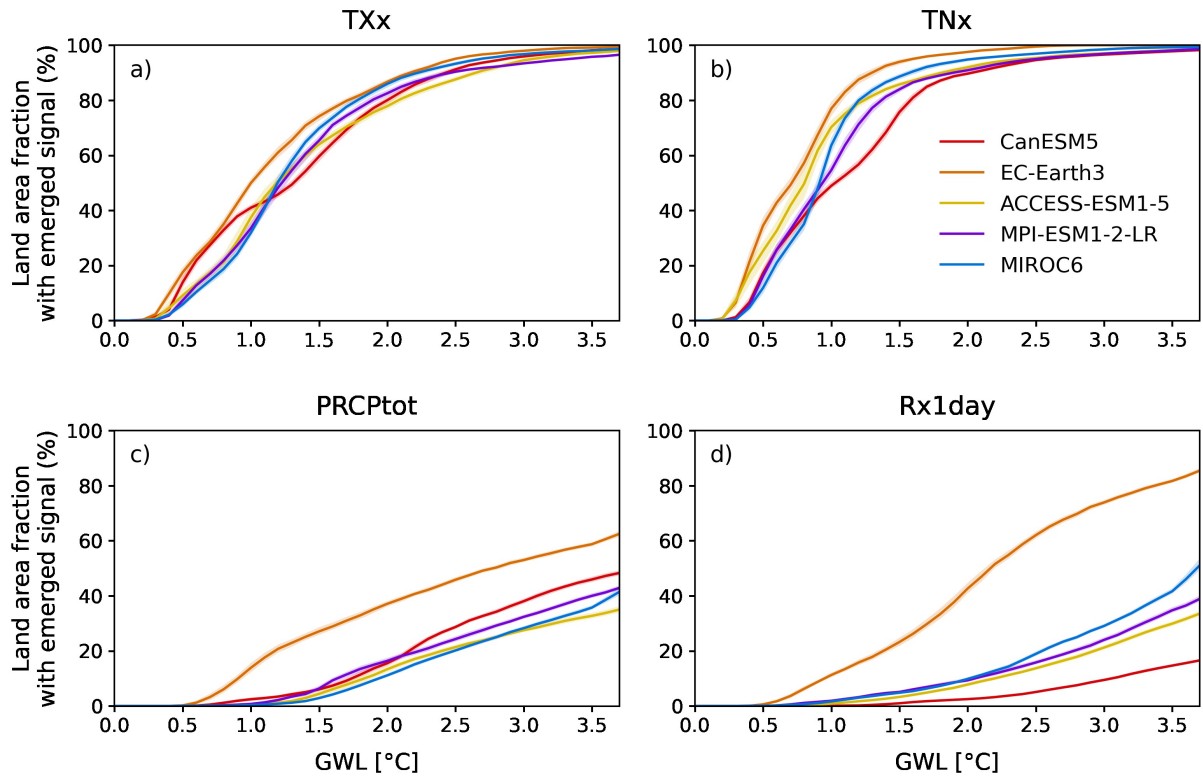

**Figure 3: Fraction of land area exposed to emerged climate indices in dependence of global warming level (GWL). The respective land area fraction is presented for emerged signals of TXx (a), TNx (b), PRCPtot (c), and Rx1day (d). Different colors represent the five applied SMILEs (with equilibrium climate sensitivity (ECS) increasing from blue to red) and shading indicates the sampling uncertainty (95% confidence interval estimated by bootstrapping, see Methods).**

We further estimate the percentage of global population that is exposed to emerged climate signals, considering the uncertainty due to different population projections according to the scenarios SSP1 to SSP5 (Fig. 4). In general, the patterns of population exposure follow the path of land area exposure, with large fractions of the global population being affected by emergences of TXx and TNx already at low GWLs. In contrast, PRCPtot and Rx1day will emerge at higher GWLs and consequently affect fewer people. For TXx, the exposure under present-day climate shows a rather large spread (affecting 37%-56% of global population) but increases to 68%-88% at a GWL of 2.0°C (Fig. 4a). Regarding TNx, already 72%-80% of the global population is exposed to emerged signals under present-day climate, with model agreement being higher than for TXx (Fig. 4b). This percentage is projected to increase to 86-93% at 1.5°C, and at 2.0°C virtually everybody (more than 95% in all five SMILEs) will be exposed to a new climate state of TNx (Fig. 4b). Under present-day warming, the highest exposure to TNx emergence can be found in Africa, Southeast Asia, South America, and North & Central America where more than four out of five people already experience an emerged climate signal (Supplementary Fig. S5). For PRCPtot, we find lower exposure where up to a GWL of 2.0°C only a small but steadily increasing fraction of the population (3%-13%) will experience a new climate state.

For Rx1day, the exposed population starts to steadily increase at a GWL of 1.0°C but remains below 13% up to a GWL of 2.0°C in four out of the five SMILES (40% in EC-Earth3). The projections of the different SMILEs diverge at higher GWLs, with EC-Earth3 showing the largest, and CanESM5 the smallest increases. Particularly pronounced increases in exposure to Rx1day are found in Africa and South America, although with large uncertainties across SMILEs (Supplementary Fig. S5).

Again, the sampling uncertainty is rather low at global scale for all four climate indices (Figure 4). Regionally, however, it represents a substantial source of uncertainty (Supplementary Figure S3), for example in Southeast Asia and Australia for all four climate indices and, for PRCPtot and Rx1day, also in several other regions. For most indices and regions, the uncertainty due to different population projections only plays a minor role and is even smaller than the sampling uncertainty. For TXx and TNx, the differences across models clearly dominate the uncertainty of the globally exposed population (Fig. 4a, b) to emerged climate signals. For PRCPtot and Rx1day, the differences across models also dominate the overall uncertainty but the spread due to the population scenarios becomes more prominent (Fig. 4c, d).

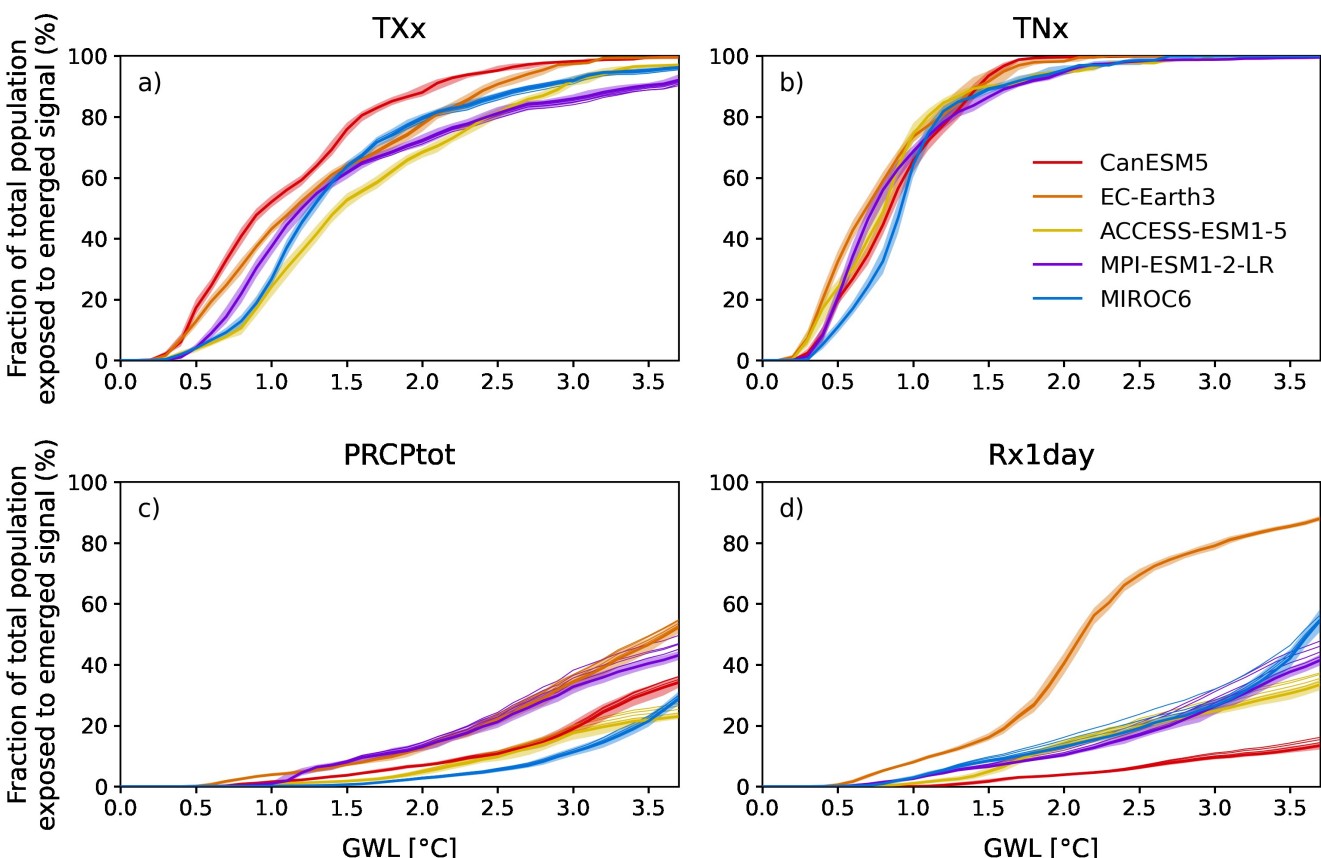

**Figure 4: Percentage of global population exposed to emerged climate indices as a function of GWL. The respective exposed population fraction is presented for emerged signals of TXx (a), TNx (b), PRCPtot (c), and Rx1day (d). Different colors represent the five applied SMILEs (with equilibrium climate sensitivity (ECS) increasing from blue to red) and shading indicates the sampling uncertainty (95% confidence interval estimated by bootstrapping see Methods). The different lines indicate the exposure according to different population scenarios, with the thick line corresponding to a population development according to SSP5 and the thin lines to population developments according to SSP1-SSP4.**

## 3.3 Increase in probability ratios for different global warming levels

Next, we investigate how the frequency of extremes in the four climate indices changes with GWL by examining differences in their probability ratios (PR) relative to pre-industrial conditions (see Methods). While for TXx and TNx, a widespread emergence of a climate change signal regarding the local full index distributions (based on a KS-test; Fig. 2) is detected for GWL < 1°C, the following analysis focuses on the tails of the distribution. Both temperature indices show a widespread increase in the frequency of extreme events (positive changes in the probability ratio) across all continents in the GWL range 1-2°C (Fig. 5). For TXx and TNx, the increase in PR is largest for central North America, South America, the Mediterranean, and central Asia, and more pronounced for TNx than TXx. In these regions, the PR increase per 0.1°C warming is larger than 1. Thus, every additional 0.1°C increase in GSAT leads to an increase in extreme event frequency by at least the number of events in the pre-industrial reference period. Furthermore, the increase in probability ratios of TXx and TNx indicates a non-linear behavior. Largest increases in probability ratios are found in the GWL range of 1-2°C, thus at a GWL at which TXx and TNx already show widespread emergence (Fig. 2). These increases get less steep once the peak of the index distribution crosses the defined threshold for extreme events (95% percentile in 1850-1900) and they stabilize towards higher GWLs (at 3-4°C or even higher). The PR patterns remain similar for a more extreme threshold (99[th] percentile, corresponding to a 100-year return period), albeit yielding higher increase rates given the lower number of events (Supplementary Fig. S7).

For PRCPtot and Rx1day, changes in PR are generally less pronounced than for TXx and TNx (Fig. 5). They increase by 0.25 to 0.75 per 0.1°C (corresponding to a 25%-75% higher probability of extreme events per 0.1°C warming) in the Northern high latitudes, Africa, the Himalaya region, and – for Rx1day – parts of South America. These regions also emerge as hotspots under a more extreme threshold (99[th] percentile; Supplementary Fig. S7). With the precipitation indices showing emergence at rather high GWLs, or no emergence at all, low PR changes in most regions (aside from the Northern high latitudes and tropical Africa with GWLoE <= 2°C) originate from slow distribution shifts where the extremeness threshold has not been crossed yet. This is contrary to TNx and TXx. For PRCPtot, several regions show a decrease in the probability ratio of -0.25 per 0.1°C warming (Central and South America, southern Africa, the Mediterranean region, and parts of Australia), indicating a general decrease of precipitation in these regions, in line with findings of the recent IPCC assessment report 6 (IPCC, 2021). Regions with decreasing PR show lower model agreement than regions with increasing PR. In contrast to the temperature indices, the change patterns of PR for PRCPtot and Rx1day remain similar across GWL ranges indicating a more linear response to changes in GSAT for the considered GWL range (1-4°C).

In several regions, PRs level off at high GWLs (Fig. 5). If this occurs in regions with early GWLoE it indicates that the maximum possible exceedance probability is reached: Each year surpasses the pre-industrial reference threshold. This GWL

of saturation is generally much lower for TXx and TNx than for PRCPtot and Rx1day (Fig.6), with saturation being reached below 2°C in South and Southeast Asia, and large parts of Africa and tropical South America. Parts of North America and northern Australia reach saturation between 2°C and 3°C (Fig. 6). In contrast, the precipitation indices (PRCPtot and Rx1day) reach saturation in much fewer grid cells and at much higher GWLs, except for the Mediterranean, and parts of South America

reaching saturation for PRCPtot. At locations and GWLs where saturation is reached, the projected index distributions show close to no overlap with pre-industrial conditions.

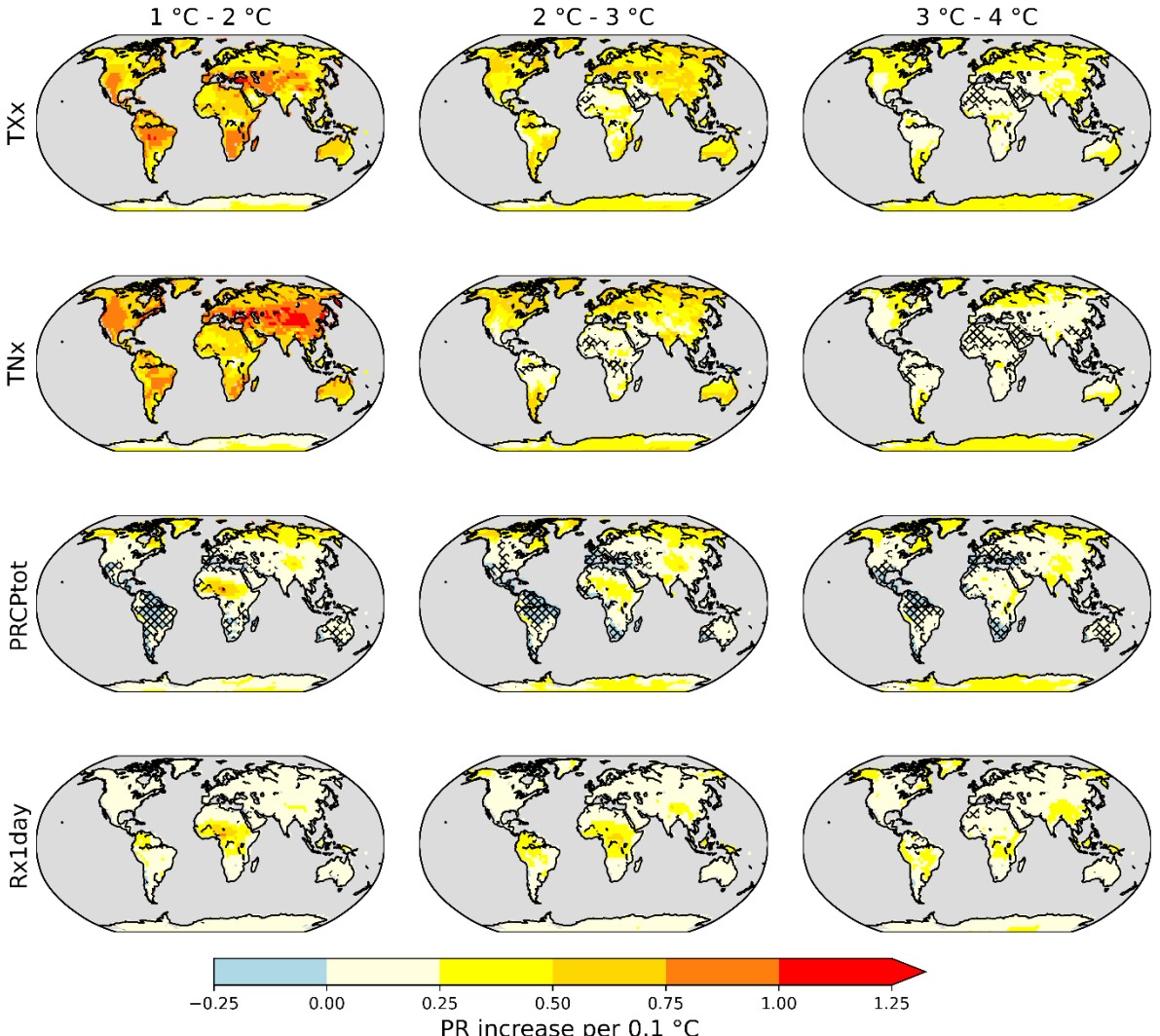

**Figure 5: Probability ratio (PR) increase for extremes in TXx, TNx, PRCPtot, and Rx1day in three ranges of global warming (1°C – 2°C, 2°C – 3°C, and 3°C – 4°C) per 0.1°C warming with respect to 1850-1900. PRs are calculated as the change in exceedance of**

**the 95th percentile of the index distribution in 1850-1900. Yellow-to-red colors indicate increasing PRs, while blue colors indicate decreasing PRs. Hatched areas indicate regions with low model agreement (at least 1 SMILE disagreeing in the sign of PR).**

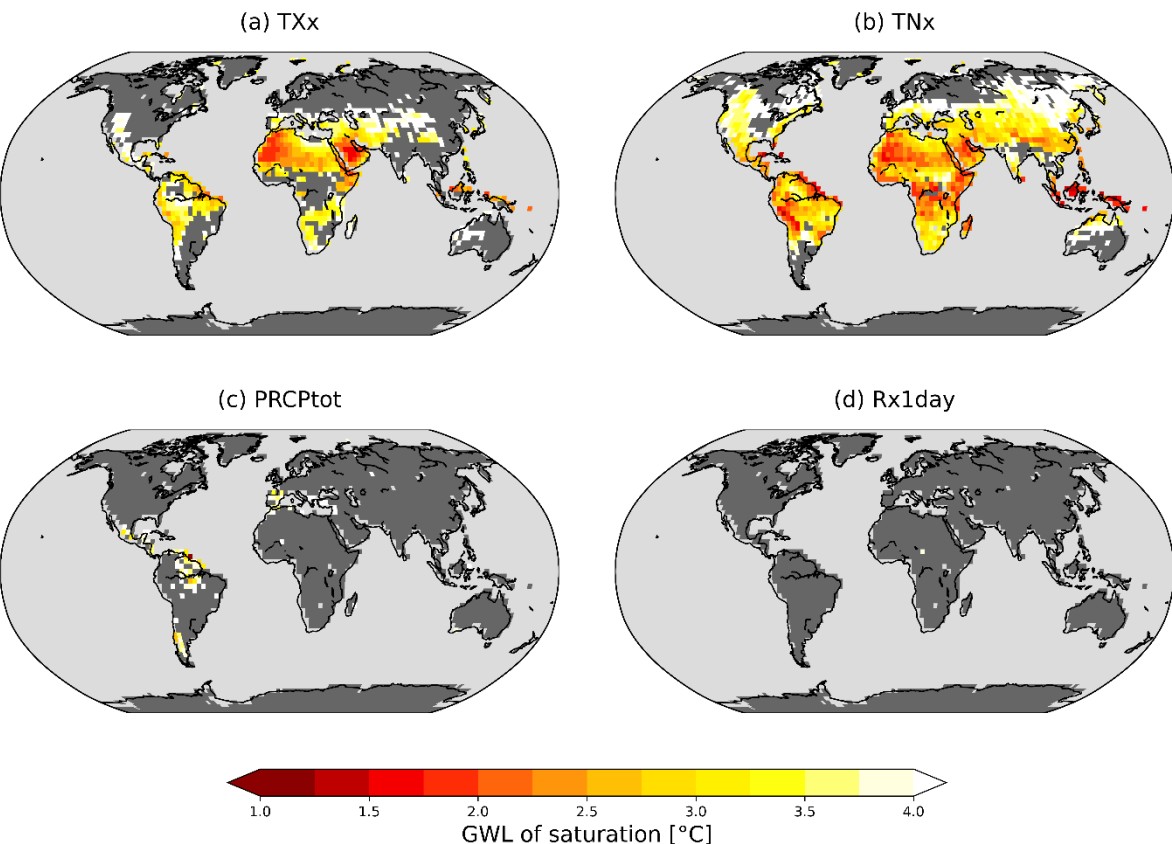

**Figure 6: Global warming level (GWL) of saturation for extremes of the selected indices. Saturation maps for TXx (a), TNx (b), PRCPtot (c), and Rx1day (d) based on values exceeding the 95th percentile of the index distribution in 1850-1900. Saturation is defined as the GWL where the maximum number of extreme events in the analysed 20-year periods is reached, i.e., each year is an extreme year relative to pre-industrial conditions. The values indicate the ensemble median across all SMILEs if at least 4 out of 5 SMILEs show saturation values. Grid cells that indicate joint saturation at GWLs higher than 4°C are colored in white, areas that show saturation in fewer than four of five SMILEs are colored in dark grey, and non-land grid cells are colored in light grey.**

## 4 Discussion

### 4.1 Adverse impacts of incremental GWL changes

Our results highlight that incremental GWL changes (i.e., steps of 0.1°C) can strongly increase the emergence of new climate states for the investigated indices. This is particularly the case for temperature extremes (TXx and TNx), for which we find widespread emergence already under present-day GWL. This finding is in line with the increasingly frequently observed heat extremes that can be attributed to climate change(Ciavarella et al. 2021, Philip et al. 2022, Philip et al. 2023). The widespread emergence of TNx under current climate conditions is of particular concern, as it corresponds to elevated nighttime temperatures. This can reduce people's recovery potential and may thus adversely impact their health conditions (Royé et al.,

2021, Thompson et al., 2022). At the same time, the precipitation indices Rx1day and PRCPtot start to emerge in the GWL range 1-2°C. This indicates that we are currently in a crucial period, where every fraction of a degree of additional warming may cause further regions to transition into new climate states, in terms of both mean and the tails of the index distributions.

Limiting global warming to 2.0°C would keep the population and land fraction exposed to emergences of Rx1day and PRCPtot below 20%. Beyond 2.0°C the exposure to emergences of these indices will rapidly increase. The current policies, which put the world on track to reach a warming of 2.8°C by 2100 (Liu & Raftery, 2021), would thus expose a considerable fraction of population and land to new precipitation regimes and most of the population and land area to new temperature regimes (Supplementary Fig. S3 & S5), potentially outside the human climate niche (Lenton et al. 2023). Additionally, the spatial

patterns of exposure rates and the frequency of future extremes show a strong regional heterogeneity, which might lead to increased socioeconomic inequality, especially in poorer regions of the world (King & Harrington, 2018).

**4.2 Non-linearities and saturation of probability ratios**

The responses of temperature and, to a lesser extent, precipitation extremes to global warming appear to follow a non-linear path (Fig. 3 & 4). However, this does not directly speak to the linear or non-linear growth of extremes. Rather, in each grid

cell the majority of the distribution of a given variable crosses the extremeness threshold at a distinct GWL (see schematic illustration in Supplementary Fig. S1a). The contribution of this grid cell to the fraction of emerged land is zero before the crossing, and equal to the fractional area of the grid cell afterwards. This continues simultaneously across all grid cells, forming the distribution of emerged grid cells in dependence of the GWL. The increase in emerged land fraction (or population) is particularly steep until the majority of the grid cells passed the threshold and consequently flattens out afterwards. Once the

thresholds are exceeded in all grid cells, the fraction of emerged grid cells reaches 100% and can no longer increase.

Our results show a very rapid initial growth (i.e., a large fraction of grid cells emerge at similar GWLs) particularly for TNx and (slightly less pronounced) for TXx, in line with saturation patterns corresponding to the non-linear growth seen for CMIP5 models (Fischer & Knutti, 2015). For precipitation, the fraction of emerged land increases less steep, in line with a more linear growth as seen also in the CMIP5 results of Fischer & Knutti (2015). The respective trajectories of precipitation and

390 temperature extremes are nevertheless alarming. First, the sharp increase of emerged temperature extremes will strongly increase the human exposure to extreme heat. Second, regional preparedness to future temperature extremes might be insufficient for unexpectedly rapid changes in the occurrence of extremes (King et al. 2018). The usage of small GWL increments (e.g., 0.1°C as used in this study) thus seems imperative, as an assessment across large increments (e.g., 0.5-1.0°C) might undersample the temperature axis and potentially mask changes in the slope of the underlying distribution. It also allows

to demonstrate that implementing further policies to reduce global warming is not futile, even if they result in incremental reductions only – which would be impossible when employing larger increments.

Probability ratios of the temperature indices increase considerably up to a GWL of 2.0°C with widespread saturation reached at a GWL of 2.0°C. This would imply unprecedented heat conditions in Southern Asia, northern Africa, and northern South America for most years even if the 2.0°C target of the Paris Agreement was met (Fig. 5). Precipitation indices reach saturation

only at higher GWLs, which points towards more inert adjustments of precipitation to changing climate. It is important to emphasize that the interpretation of saturation levels (which are reached in widespread regions particularly for temperature indices) should not be overly generalized. They are subject to the considered index and the underlying distributions and dependent on the applied threshold (here 95[th] percentile; see Supplementary Figs. S8-S10 for other percentiles) and the defined reference period (here pre-industrial conditions; Harrington & Otto, 2018). Considering this, they can be used as a tool to indicate that events considered "extreme" under pre-industrial conditions occur on a yearly basis once saturation occurs and thus become the new normal state. Reaching the saturation level of exceedance, however, should not be confused with reaching a 'safe' state and does not impede further changes in the magnitude and intensity of extremes (Harrington & Otto, 2018). Instead, the exceedance of greater extremes (i.e., higher percentile thresholds) likely continues to rise. Even hotter temperatures and heavier precipitation events are expected to occur at higher GWLs (Supplementary Fig. S1a).

## 4.3. Dependence of climate signal emergence on remapping sequencing

To combine and display climate data with different spatial resolution, remapping is essential. However, the order of operations may vary, targeted towards the specific scope of the study. Here, we remap the data to the grid of the coarsest model (CanESM5) *after* calculating the climate indices (TXx, TNx, PRCPtot, Rx1day) on their native resolution. This sequencing takes advantage of model diversity by preserving the precipitation and temperature fields of the models with higher spatial resolution when calculating the indices. It yields a local representation of the considered extreme indices, similar to what observational data sets would deliver (de Vries et al. 2023). Alternatively, climate data can be remapped *before* calculating the climate indices. This sequencing would lead to more harmonized model results but removes the fine scale information provided by models with higher spatial resolution. For studies analyzing model performance and focusing on model comparison, the latter approach would be preferable.

The impact of the processing order on the resulting fields is expected to be more substantial for daily precipitation extremes (such as Rx1day) than for temperature or total annual precipitation. When these precipitation extremes are calculated on the finer grid, they might occur on different days and would then be aggregated to form the larger grid cells of the remapped data. Remapping *before* the calculation of the extreme indices, would keep the temporal integrity but results in a dilution of the precipitation extremes that often occur more locally.

For our study, the former approach (remapping *after* calculating the indices) is advantageous, as we aim to investigate local emergences of climate change signals and the related exposure of population. Moreover, we focus on relative changes in the indices (assessed via ToE, GWLoE, PR) rather than changes in their absolute values. We find only negligible difference between both remapping orders for TXx, TNx and PRCPtot for the land area fraction exposed to emerged signals (Supplementary Fig. S11 & S12). However, we identify a substantial divergence for the emergence of Rx1day. Focusing on local level extremes (remapping *after* calculating Rx1day) yields earlier Rx1day emergences compared to the approach that harmonizes model results (remapping *before* calculating Rx1day). Additionally, the latter approach reduces the model spread

in case Rx1day emergences are expressed as function of GWL (Supplementary Fig S12). This spread remains unchanged if emergences are expressed as a function of time (Supplementary Fig. S11). This indicates that most of the model spread for Rx1day emergences expressed as function of GWL can be explained by model resolution, whereas the different ECS seems to play a secondary role (Fig. 3d, Supplementary Fig. S12d, Tab. 1). The high sensitivity of ToE/GWLoE to the selected remapping order for Rx1day (and presumably also for similar precipitation indices) highlights that this sequencing is of great importance for quantifying related emergences. The decision on performing the remapping *before* or *after* calculating the desired index should thus not be an arbitrary choice. Our results highlight that this is crucial not only for the investigation of changes in absolute values but also when ToE or GWLoE are of interest.

**4.4 The concept of GWLoE as a tool to communicate climate change impacts and related uncertainties**

Combining the concept of ToE with global warming levels supports a more policy-relevant communication of the emergence of climate signals as global policies are more frequently based on warming levels (e.g., 1.5 or 2.0°C targets of the Paris Agreement). We find that GWLoE provides a feasible tool to constrain model uncertainty, particularly for temperature variables and temperature-related indices. We generally find a higher model agreement for TNx and TXx if emergence is expressed as a function of GWL (Supplementary Fig. S3, S5, S13) instead of time (Supplementary Fig. S4, S6, S14). However, regional differences remain. For PRCPtot and Rx1day, in contrast, we find better agreement across SMILEs when expressing emergence as a function of time. This indicates that precipitation changes are not only impacted by thermodynamics but also by other processes, such as aerosol forcing (Lin et al. 2016; Lehner & Coats 2021), which are characterized as a function of time rather than GWL. In that regard, precipitation changes are more dependent on the scenario pathway and thus more prone to scenario uncertainties in some regions (Maher et al. 2019). Additionally, precipitation changes are more affected by small-scale processes and thus model resolution, which contributes to the larger model spread for precipitation than temperature indices as discussed above.

Internal climate variability represents a major source of uncertainty for the estimation of GWLoE and thus needs to be accounted for. Across indices, the spread originating from internal variability (Supplementary Fig. S15) is of similar magnitude as the inter-model range across SMILEs (Fig. 2). For the temperature indices, internal variability regionally even exceeds the across-model spread, particularly towards the poles (Supplementary Fig. S15). This becomes even more relevant for the assessment of impacts at low GWLs, i.e., projections for the upcoming decades, where internal climate variability is a particularly large source of uncertainty (Hawkins and Sutton 2009, Lehner et al. 2020). This makes SMILEs an essential tool to determine GWLoE as they allow to quantify internal variability and thus derive a robust signal detection even at low GWLs (Maher et al. 2020). To further increase the robustness of the GWLoE estimates, we apply a 90% emergence threshold across members (see Methods). This rather conservative estimate ensures internal variability is properly accounted for. Considering the joint emergence of multiple SMILEs then further increases the robustness of GWLoE estimates and constrains both internal variability and model uncertainty across a wide range of GWLs.

Further, the approach considering GWL rather than time to estimate emergence might be beneficial to overcome the "hot model problem" (Hausfather et al. 2022), i.e., the issue of selecting climate models that show a higher-than-average equilibrium climate sensitivity (ECS) to increasing $CO_2$ levels (Suarez-Gutierrez et al. 2021). We find that a time-dependent approach will generally lead to a model order, where models with high ECS (Tab. 1) usually show the highest exposure of population and land area to emerged climate signals (Supplementary Fig. S4, S6, S14). In contrast, our results show that a GWL-centered analysis results in a model order that is largely independent of the models' ECS (Supplementary Fig. S3, S5, S13). This holds particularly true for temperature indices and to a lesser degree also for PRCPtot and Rx1day. In particular, for Rx1day model resolution seems to be more impactful than ESC.

Our results are presented for GWLs extracted from simulations of transient climate that do not necessarily comply with equilibrium conditions with long-term stabilization at a certain GWL (Mitchell et al. 2016). Regional warming in model experiments with quasi-equilibrium climate states can be expected to be cooler than in transient warming scenarios (King et al. 2020). This becomes even more prominent for the magnitude of summer extremes, in turn also affecting their frequency (King et al. 2020). Consequently, our results do not reflect stabilized climate states as, for instance, occurring in overshoot scenarios, and should thus not be misinterpreted as long-term impacts if specific GWLs were met (e.g., the 1.5 °C target). Quantifying emergences under equilibrium conditions would require a different study design with an ensemble of SMILEs with stabilized GWLs (Mitchell et al. 2016, King et al. 2020). Instead, our results represent snapshots of GWLs under transient climate conditions with focus on their dynamics and changes at incremental GWLs, which remain valid under the given constraints.

Finally, our results are based on the high-end warming scenario SSP5-8.5, which is considered to project a low-probability high warming for the end of the 21st century, given current climate policies (Hausfather & Peters, 2020). Analyzing the impacts of high warming levels (>3.0°C), however, requires the selection of rather extreme warming scenarios (SSP3-7.0 or SSP5-8.5), as these scenarios are the only ones that reach sufficiently high warming (Meinshausen et al. 2020). Furthermore, temperature and precipitation changes were found to scale largely linearly across scenarios for moderate GWLs (Seneviratne et al. 2016). Given that we use a cut-off GWL of 4°C, our results can still be considered robust for the range of GWLs that we investigate.

**5 Conclusions**

In this study, we present the global warming level of emergence (GWLoE) of four temperature and precipitation indices (TXx, TNx, PRCPtot, and Rx1day) and the related exposure of population and land area based on the joint emergence of five SMILEs. Under current warming levels, large parts of the global population and global land area are already exposed to TXx and TNx emergences, while PRCPtot and Rx1day are about to emerge in several regions. We find widespread emergence of TXx and TNx at a GWL of 2.0°C and mostly linear increases in the emergence of PRCPtot and Rx1day over the GWL range 1.0-2.0°C.

Emergences of TXx, PRCPtot, and Rx1day continue increasing beyond 2.0°C. These results confirm that a GWL of 2.0°C should not be misinterpreted as a safe target (Knutti et al. 2016). For higher warming levels (>2.0°C) strong increases in the fraction of exposed land area and population to emerged climate signals were identified for precipitation indices (PRCPtot and Rx1day). Further, we identify a sharp increase in the frequency of temperature extremes (assessed through probability ratios of TXx and TNx), particularly at lower GWLs. These results highlight that considering incremental GWL steps for analyzing the emergence of climate change signals is essential.

Given the dominant role of internal variability at low GWLs that are close to present-day warming, we argue that large ensemble simulations are essential: first, to robustly detect the emergence of climate change signals and second, for their assessment at incremental GWL steps, particularly for analyses of extreme events, which we find to be of highly relevant magnitude at low GWLs. Using GWLs over time to detect the emergence of climate change signals proves to be particularly well suited for temperature-based indices, as it substantially reduces the uncertainty of signal emergence compared to a time-based approach. For precipitation-based indices, we find lower uncertainties when expressing their emergence as a function of time instead of GWL. The decision of whether to apply GWLoE or ToE thus depends on the considered climate variable. Additionally, regional specifications should be respected, as indicated by the large regional discrepancy in our results. Further, the strong sensitivity of the emergence of Rx1day on the remapping sequencing highlights the need to tailor the order of remapping to the individual research focus of each study.

Our results underline the importance of climate mitigation and the imminent need for an early achievement of net-zero emissions (Iyer et al. 2022) to avoid strongly increasing emergences of temperature and precipitation indices. This urges for the implementation of policies to ensure that global warming is limited at least to the targets defined in the Paris agreement. Every fraction of a degree matters to prevent additionally emerging adverse effects of climate change on human wellbeing.

**Code and Data Availability**

The applied CMIP6 SMILEs with the corresponding variables and projections under historical and SSP5-8.5 scenarios are available under the ESGF nodes (e.g., https://esgf-data.dkrz.de/projects/esgf-dkrz/). Population data can be accessed via https://www.isimip.org/gettingstarted/input-data-bias-adjustment/. The codes and data to derive the respective figures are shared via a Figshare repository (https://doi.org/10.6084/m9.figshare.25428100.v1).

**Author Contributions**

While the presented study originates from a joint hackathon meeting and thus represents a true group effort, the following individual contributions are recognized. D.G. and R.R.W developed the general study concept and all authors contributed to the detailed outline of the study. D.G. contributed with the calculation of the joint GWLoE and drafted the first version of the manuscript. C.S. provided the GWLs for all SMILEs, performed the exposure analysis, drafted the corresponding section in

the manuscript and provided the related figures. A.B. calculated the probability ratios, drafted the corresponding sections in the manuscript, and produced the presented maps. M.M. led the SMILE selection process, contributed with coding support and conducted a critical literature screening. M.S. provided the ETCCDI precipitation and temperature indices. All authors actively discussed all results and contributed extensively to writing the final manuscript.

## Competing interests

The authors declare that they have no conflict of interest.

## Acknowledgements

We further acknowledge the World Climate Research Programme, which, through its Working Group on Coupled Modelling, coordinated and promoted CMIP6. We thank the climate modeling groups for producing and making available their model output, the Earth System Grid Federation (ESGF) for archiving the data and providing access, and the multiple funding agencies who support CMIP6 and ESGF.

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
