# Peer review of "Applying global warming levels of emergence to highlight the increasing population exposure to temperature and precipitation extremes"

_EGUsphere, 2023_

## Author Response (AR1)

**Reply to reviewers for Gampe et al. (2024) submitted to ESD**

The authors sincerely thank all three anonymous reviewers for their time and dedication to improve our manuscript and providing valuable feedback to us. We revised the submission accordingly. Please find the specific replies to the individual comments below. The original comments are stated in italic and the reply is written in red color. All text passages are copied from the revised manuscript, line numbers refer to the tracked-changes version of the revision for easier findability.

**Reply open discussion Reviewer 1**

Thank you for the valuable inputs and comments and for your time and effort to critically review our manuscript. We followed your suggestions to improve our manuscript.

1. *A benefit of SMILEs is that they can be used to explore sampling as well as structural uncertainties. I think with Figure 3 in particular it would be useful to show a range of area of emergence as a function of GWL for each SMILE. This could be derived from bootstrapping the simulations and computing a confidence interval based on the resampled ensembles.*

We highly appreciate your suggestion to use bootstrapping to derive an uncertainty estimation of sampling the GWLs. Consequently, we bootstrapped the selected members for each SMILE (1000 repetitions with replacement). We opted not to reduce the number of members selected in each of the 1000 samples in order to maintain the critical number of 30-50 members throughout the process. We checked to ensure an equal distribution among selected members. The resulting sampling uncertainty is included in the respective figures via the 95% confidence interval and both, methods and discussion section have been expanded accordingly.

L.153-159: To estimate the sampling uncertainty in the calculation of the emergence, a bootstrapping approach is conducted. We sample $n$ members ($n$ = ensemble size) of each SMILE from the available members with replacement, applying 1000 repetitions. Thereby, all individual members were sampled approximately equal times. For each of these 1000 bootstrapped ensembles, we then assign ToE to the year when at least 90% of the drawn ensemble members show an emerged climate signal (e.g., 45 of 50 members; similar to the approach by Martel et al. 2018). This method accounts for internal variability, expressed via the $n$ SMILE members. As we require 90% of the members to be emerged, the approach yields a conservative yet robust estimate of ToE. The sampling uncertainty is presented as the 95% confidence interval derived from the 1000 ToE estimates.

2. *Some SSP population projections aren't particularly compatible with some emissions pathways. As such, SSP1 population projection is unlikely to be compatible with having a high GWL. I would suggest that SSP5 populations are used in Figure 4 and that uncertainty estimation through bootstrapping (as discussed in the previous comment) is shown instead.*

Thank you for this relevant comment. We agree that not all SSPs are compatible with the selected SSP585 scenario. Nevertheless, we argue that one advantage of the application of GWLs is constraining scenario uncertainties. While the selection of the appropriate SSP is certainly a prerequisite we decided to include different spatially explicit estimates of future population distribution here to indicate the related uncertainties. Interestingly, the related uncertainties largely fall within the sampling uncertainty (comment 1). In our opinion, including the different population estimates provided additional insights. However, we clarified the potential mismatch in the respective sections to avoid any confusions.

3. *The results shown are applicable to the climate under a very high rate of global warming, but it should be noted that they aren't applicable to slower warming or stabilised climate states (e.g.*

*King et al. 2020 (https://www.nature.com/articles/s41558-019-0658-7) and Mitchell et al. 2016 (https://www.nature.com/articles/nclimate3055)).*

Thank you for bringing this to our attention and highlighting this important aspect. We agree, it is definitely important to clarify this. We thus added a new section to the discussion (l. 472-481) to address this:

l. 472 -481: Our results are presented for GWLs extracted from simulations of transient climate that do not necessarily comply with equilibrium conditions with long-term stabilization at a certain GWL (Mitchell et al. 2016). Regional warming in model experiments with quasi-equilibrium climate states can be expected to be cooler than in transient warming scenarios (King et al. 2020). This becomes even more prominent for the magnitude of summer extremes, in turn also affecting their frequency (King et al. 2020). Consequently, our results do not reflect stabilized climate states as, for instance, occurring in overshoot scenarios, and should thus not be misinterpreted as long-term impacts if specific GWLs were met (e.g., the 1.5 °C target). Quantifying emergences under equilibrium conditions would require a different study design with an ensemble of SMILEs with stabilized GWLs (Mitchell et al. 2016, King et al. 2020). Instead, our results represent snapshots of GWLs under transient climate conditions with focus on their dynamics and changes at incremental GWLs, which remain valid under the given constraints.

Minor comments:

*Figure 1: Could you check that the lines are plotted correctly. For some models the range appears considerably smaller than due to interannual variability alone in the observations. In Maher et al. 2019 (https://agupubs.onlinelibrary.wiley.com/doi/full/10.1029/2019MS001639), the MPI range looks larger than is plotted here.*

Thank you for critically reviewing this Figure. The GWLs for each ensemble were calculated using a 20-year moving window. This ultimately leads to a smoothening for individual years thus reducing the overall range considerably. As we then apply the GWLs based on the 20-yr window for further calculation of our results we decided to only show these in Fig.1. We edited the figure caption accordingly to clarify this and avoid misinterpretations.

*L190: There's a strange space that should be removed.*

Removed, thank you.

*L200: "less" should be "fewer"*

We found several other occurrences and revised all of them, thank you.

**Reply open discussion Reviewer 2:**

Major Comments:

1. *The abstract needs to be reformatted. In the current version, the authors introduce the concept of ToE and GWLs, and mention that 'ToE and GWL have barely been combined so far'. Yet, the scientific question of the research is not clear. The authors should clearly crystalize the specific problem/knowledge gap the study is aiming to address, i.e., the shortcomings of current research and why combining ToE and GWLs is significant.*

Thank you for this valuable comment that led to a thorough revision of the first portion of the abstract. We focused on the importance of GWLoE and argue that our approach provides a robust estimation of GWLoE (using multiple SMILEs).

l.14-21: The swift and ongoing rise of global temperatures over the past decades led to an increasing number of climate variables showing statistically significant changes compared to their pre-industrial state. Determining when these climate signals emerge from the noise of internal climate variability (i.e., estimating the Time of Emergence, ToE) is crucial for climate risk assessments and adaptation planning. However, robustly disentangling the climate signal from internal variability represents a challenging task. While climate projections are communicated increasingly frequently through global warming levels (GWLs), ToE is usually still expressed in terms of time horizons. Here, we present a framework to robustly derive global warming levels of emergence (GWLoE) using five Single Model Initial-condition Large Ensembles (SMILEs) and apply it to four selected temperature and precipitation indices.

2. *Again, there is a need to revise the Introduction to make it more accessible, particularly the current research progress and the knowledge gap. After reading the introduction, I am not clear about the current research progress and still have doubts about the importance of the paper. The authors only mention that "Recently, first studies combined GWL and ToE to provide global warming levels of emergence (GWLoE) instead of ToE (Abatzoglou et al. 2019, Kirchmeier-Young et al. 2019, Raymond et al. 2020). Yet, GWLoE remains a rarely applied concept in general as well as in the context of using SMILEs in particular". The readers should be aware of the knowledge gap and be convinced that it is important to fill the knowledge gap after finishing reading the Introduction.*

Thank you for critically reviewing the introduction and bringing this to our attention. We agree with the reviewer that the first version was not precise enough and lacked clearness over all. We thus revised the introduction and expanded certain sections. We highlighted the importance to account for internal variability when estimating ToE (that is largely only possible via SMILEs in that context), the transition to GWLoE and further, the use of multiple SMILEs to account for both, internal as well as structural uncertainty. The framework we propose in this study overcomes current literature shortages by accounting for all of these uncertainty sources and thus providing robust estimates of GWLoE (e.g., as joint emergence). While we overhauled the introduction at several points, we partularly highlighted the shortcomings and role of internal variability:

L 64-74: Most ToE studies use multi-model ensembles, which mainly consist of single realizations of different models, thus accounting for structural uncertainty and scenario uncertainty (Giorgi and Bi 2009, King et al. 2015, Bador et al 2016, Douglas et al. 2022). However, these single realization ensembles can only partially account for the intrinsic uncertainty due to internal climate variability. Especially on regional-to-local scales, internal variability is large compared to the other sources of uncertainty, showing the largest fractional uncertainty (Lehner et al. 2020, Blanusa et al. 2023). Accounting for internal variability when estimating ToE, is relevant since it can advance or delay ToE by up to several decades (Hawkins et al 2014), and can contribute half to two-thirds to the total ToE uncertainty (Bador et al 2016). To account for the influence of internal variability in ToE studies, Single Model initial condition large ensembles (SMILEs) can be applied. SMILEs constitute numerous independent, yet equally probable, climate simulations, created by running a single climate model

multiple times under the same external forcing (e.g., same emission scenario) but with marginally changed initial conditions (Kay et al. 2015, Maher et al. 2019).

> 3. *As for the climate indices, why choose these four of different types? The temperature indices (TXx and TNx) are absolute indices; precipitation index PRCPtot is one of the amount indices, and R1Xday is the intensity index. Why not choose the indices of the same group for both temperature and precipitation?*

This is a very valid point, thank you for highlighting the need to clarify this in our manuscript. In fact, we decided to use indices that are widely used and represent both, temperature and precipitation. This was due to the fact, that we expected the temperature indices to scale well with GSAT while for precipitation other factors that are scenario and thus time dependent (e.g., aerosols) are more crucial. It was thus essential to consider both variables for the purposes of this study. Further, we agree thtat the indices cover different types of metrics. However, this was intended to apply to concept of GWLoE to a wider range of indices. We added to the methods section to briefly mention why those were selected.

L. 115-118: We selected those four indices as they are frequently applied (e.g. Sillmann et al. 2013, Deng et al. 2022) and allow for easy interpretability. We further aim to demonstrate the concept of GWLoE for a broad range of indices. The selected temperature and precipitation indices thus intentionally cover both, absolute (TXx, TNx and PRCPtot) and intensity (Rx1day) metrics.

> 4. *Why resample to the coarsest resolution (2.8°, CanESM5)? To what extent the selection of the resample resolution affects the exposure of land area and population? This should be clarified.*

Thank you for this valid and important comment. One of the foci of our study is the joint emergence using multiple SMILEs to increase the robustness of our results (considering also structural uncertainty) on a grid scale. This, however, requires to harmonize the SMILEs to a common grid as otherwise coarser models would be penalized for their resolution, likely skewing the results. This step is in our opinion crucial, and thus acknowledged and highlighted in the discussion and the supplementary figures. However, we agree that our reasoning to resample was lacking in our manuscript at the current stage and revised the corresponding methods section accordingly:

L. 119-125: To make the results comparable across SMILEs and to further calculate the joint emergence using multiple SMILEs, the grids of the five SMILEs must be harmonized. Typically, either the finest or coarsest grid is selected as target resolution. The selection of the finest grid exploits the potential of the high-resolution models. The coarser models, however, might not be capable of resolving the processes at the higher resolution for structural and parameterizational reasons (Prein et al. 2016). Using the finest grid would then also require the introduction of new data points (either through interpolation or downscaling). We thus opted to remap all data sets to the spatial resolution of the coarsest grid (CanESM5, ~2.8°x2.8°; Tab. 1) using a first order conservative remapping approach.

> 5. *Figure 1, the colored lines are very smooth, indicating almost no inter-annual variability (IAV). Has the time series been smoothed? The author should explain that.*

Thank you for critically reviewing this Figure. The GWLs for each ensemble were calculated using a 20-year moving window. This ultimately leads to a smoothening for individual years thus reducing the overall range considerably. As we then apply the GWLs based on the 20-yr window for further calculation of our results we decided to only show these in Fig.1. We edited the figure caption accordingly to clarify this and avoid misinterpretations.

> 6. *Figure 2, considering that there is almost no visible spatial difference in Figure b, the color bar can use unequal spacing instead of equal spacing.*

We agree there is only little spatial difference in Fig. 2b. However, we propose to keep the equal spacing for better comparability with the other indices. As the emergence for TNx has largely already occurred at present-day GWL (1.1°) only little added value would be gained (and the comparability lost).

*Specific Comments:*

1. *Many sentences lack commas, making them difficult to read. For example, Line 54, a comma is needed between "To disentangle a robust climate change signal from the background noise of internal climate variability" and "Single Model Initial-condition Large Ensembles (SMILEs) are widely used (e.g., Deser et al. 2020, Maher et al. 2021)".*

We appreciate the comment. We revised the entire manuscript and improved it also from a language point.

2. *Line 45. 'W/m2' ->' W/m$^2$'.*

Revised accordingly, thank you.

3. *Line 422. 'proofs' ->' proves'.*

Revised, thank you for catching that!

**Reply open discussion Reviewer 3:**

1. *What is the motivation to use an ensemble of SMILEs? The reason for using a SMILE is that it eliminates uncertainty that stems from using different models yet retains the internal variability (as the authors mention several times in the paper). However, combining several SMILEs again introduces uncertainty from differences between the different model, and you give away the main advantage of a SMILE. Could you comment on that?*

Thank you for this comment and for highlighting this important aspect. There is a need to use SMILEs for the purpose of ToE/GWLoE estimation in order to properly account for internal climate variability. This, however, comes at the "cost" of neglecting structural uncertainty (in case only 1 SMILE is used). We aimed to provide a very robust framework to estimate GWLoE that accounts for both sources. We thus used the individual SMILEs to derive ToE/GWLoE (accounting for internal variability) and then merge them to yield the joint emergence (that also addresses structural uncertainty). We added a new figure to the supplement that quantifies internal variability and revised the manuscript at various sections to highlight that aspect. In particular, the introduction is now revised to better highlight the need for multiple SMILEs. We added to the introduction to also highlight that aspect.

l. 97-99: By using an ensemble of multiple SMILEs, we are able to robustly determine the emergence as a function of GWL at the grid scale level, implicitly accounting for internal variability and structural uncertainty. Expressing emergence as GWLs instead of time thereby constrains the scenario uncertainty.

2. *The paper hasn't convinced me that an ensemble of SMILEs is essential for this type of analysis. How different would the results have been if an ensemble had been created from all CMIP6 models instead, say 1 member from each model that contributed to CMIP6? Would this change the finding? Or the robustness?*

The main motivation of our study is to determine when a clear climate change signal emerges from the "noise". It is thus essential to separate the climate signal from internal climate variability. We calculate the emergence of such signal in each member of each SMILE using a KS-test to compare the distribution of the desired time window with the pre-industrial state. Once statistical significance is reached and maintained until the end of the time series (2100) we conclude an emergence of the signal. This would be possible also with a multi-model CMIP6 ensemble. However, with one single member or respective CMIP6 model realization, the forced response cannot be cleanly separated from the internal variability of the climate. It rather just represents just one of many possible realizations. As the main focus of SMILEs is to allow for a robust representation and thus quantification of internal variability, it makes their application essential for our study purposes. We thus apply a 90% threshold for each SMILE where at least 90% of the members have to indicate emerged signals. This additional step ensures the consideration of internal variability and increases the robustness of our results. Also, all SMILEs included are also part of CMIP6. While they do not reflect the entire structural uncertainty of the CMIP 6 archive though, their GWL spread is of similar magnitude (l. 133). We revised this section slightly to clarify that.

3. *The paper lacks a proper assessment of the robustness of the results. How robust are the estimates for the fraction of land or fraction of population that experience a change in climate indicators? Ensembles in general are very useful for quantifying confidence intervals or estimating uncertainty, so an ensemble of large ensembles should be even better to assess uncertainties, or?*

We agree that a more rigorous assessment and presentation of uncertainties is required and was lacking in the first version of the manuscript. We thus followed the suggestion by reviewer 1 to conduct bootstrapping for the sampling of GWL and ToE. The resulting updated exposure figures thus include the 95% confidence intervals to reflect the sampling uncertainty of the members. We also added a new

figure to the supplement (S15) that addresses internal variability and serves as a counterpart to Supplementary Figure S2 (inter-model spread; structural uncertainty). We expanded the discussion section accordingly to better account for the role of internal variability.

L. 509-520: Internal climate variability represents a major source of uncertainty for the estimation of GWLoE and thus needs to be accounted for. Across indices, the spread originating from internal variability (Supplementary Fig. S15) is of similar magnitude as the inter-model range across SMILEs (Fig. 2). For the temperature indices, internal variability regionally even exceeds the across-model spread, particularly towards the poles (Supplementary Fig. S15). This becomes even more relevant for the assessment of impacts at low GWLs, i.e., projections for the upcoming decades, where internal climate variability is a particularly large source of uncertainty (Hawkins and Sutton 2009, Lehner et al. 2020). This makes SMILEs an essential tool to determine GWLoE as they allow to quantify internal variability and thus derive a robust signal detection even at low GWLs (Maher et al. 2020). To further increase the robustness of the GWLoE estimates, we apply a 90% emergence threshold across members (see Methods). This rather conservative estimate ensures internal variability is properly accounted for. Considering the joint emergence of multiple SMILEs then further increases the robustness of GWLoE estimates and constrains both internal variability and model uncertainty across a wide range of GWLs.

4. *Sec 2.2 describes how ToE and GWL are computed. ToE and GWL are first computed individually for each ensemble member. Then the ToE for each SMILE is defined as the temperature at which 90% of the members show a significant change (BTW: how sensitive are results to this threshold?), while the GWL for each SMILE is the ensemble mean of the GWL from the members. So far I agree with the authors (and admit they describe the method much better than what I did here) but then this ensemble mean GWL is used to define GWLoE. However, isn't this last step removing a large portion of the sensitivity to the global warming as the temperature variability across the ensemble is completely removed. An alternative could be to compute GWLoE for each ensemble member Individually and then average these individual GWLoE over the ensemble instead. Apart from better accounting for the temperature variability across the ensemble members this would also make it rather straightforward to form large ensembles by combining realisations from different models (or combining the SMILEs into one large superensemble) and use bootstrapping for estimating uncertainties.*

Thank you for bringing this to our attention. Our original intention was to apply GWL as the forced response of each ensemble (that is, the respective ensemble mean). However, we agree with the reviewer that the calculation of GWLoE based on each ensemble member is much more straight forward and also more appropriate. We thus updated the methodology accordingly and re-ran all relevant scripts. Regarding the 90% threshold: this was selected (also following established Literature) so that the majority of members have to yield an emerged climate signal. A higher threshold, e.g., 95% would, however, put too much emphasis on the rather extreme members: for the MPI SMILE the current approach requires 27 of 30 members to show emergence. Raising the threshold to 95 would then consider 29 of 30 already. We also implemented a bootstrapping procedure to provide an estimate of sampling uncertainty of the selected members for each SMILE (1000 repetitions with replacement). We did not to modify the number of members selected in each of the 1000 samples in order to maintain the critical number of 30-50 members throughout the process. We checked to ensure an equal distribution among selected members was provided. The resulting sampling uncertainty is included in the respective figures via the 95% confidence interval and both, methods and discussion section have been expanded accordingly.

L. 153-159: To estimate the sampling uncertainty in the calculation of the emergence, a bootstrapping approach is conducted. We sample $n$ members ($n$ = ensemble size) of each SMILE from the available members with replacement, applying 1000 repetitions. Thereby, all individual members were sampled approximately equal times. For each of these 1000 bootstrapped ensembles, we then assign ToE to the year when at least 90% of the drawn ensemble members show an emerged climate signal (e.g., 45 of 50 members; similar to the approach by Martel et al. 2018). This method accounts for internal variability, expressed via the $n$ SMILE members. As we require 90% of the members to be emerged, the approach

yields a conservative yet robust estimate of ToE. The sampling uncertainty is presented as the 95% confidence interval derived from the 1000 ToE estimates.

5. *What is the motivation for defining ratios of probabilities, trends and saturation in the ratios in Sec 2.4? Sure, probabilities for passing thresholds will change with increasing temperatures, but it is unclear to me what the added value of probability ratios and their level of saturation really is. Maybe the authors could better motivate the choice of the chosen method?*

We agree that the motivation for the selection of the probability ratios was lacking in the first of the manuscript. We thus expanded the section to clarify this. In general, we focus on the assessment of the emergence of climate signals in relation to GWLs throughout the manuscript. However, we consider shifts in the entire distribution (KS-test) only, i.e., changes in magnitude/intensity and variability. Due to an overlay of mean shifts and variability changes, the number of years exceeding extreme thresholds (e.g., the 95$^{th}$ percentile, corresponding to 1-in-20-year events) defined for the reference period increase as well. In our case this refers to extremely high daily maximum and nighttime temperatures which may have detrimental impacts on human health, as well as to high annual and extreme daily precipitation sums. The exposure to extremes is of particular interest for various reasons (public health, urban planning, general adaptation measures etc.). It is thus important in our opinion to also focus on the tails of the index distribution, particularly in the scope of an exposure study. In this section, we thus aim to address potential changes in the tails of the distribution as well as changes in frequency of extremes. We agree that multiple other options for this would be available. We selected changes in PR as most appropriate to connect with the exposure of population in the previous section. Further, the PR ratio is a common framework in detection and attribution which is easy to understand and communicate: the GWL-dependent frequency of events exceeding a threshold in relation to their frequency during a reference period. The assessment of PR and related saturation in our opinion complements the other sections in a unique way. This analysis clearly highlights the need to include incremental changes compared to fixed warming levels, due to the shown non-linear response as well as the sensitivity of small incremental changes in GWL. In addition, it provides insights in extreme year frequency changes at low GWLs. Further, we argue that our setup using five SMILEs is ideal to perform this kind of analysis that expands the manuscript to also reflect extremes to some extent.

---

## Referee Report (RR1)

I'd like to thank the authors for thoroughly discussing and addressing the questions that I had to the first version of the manuscript. I'm satisfied with the author's replies and would therefore recommend the editor to accept the manuscript for publication.